# From Online Aggression to Offline Silence: A Longitudinal Examination of Bullying Victimization, Dark Triad Traits, and Cyberbullying

**DOI:** 10.3390/bs15111583

**Published:** 2025-11-18

**Authors:** Shaojie Zhang, Jiaxiang Wang, Xiong Gan, Junwei Pu

**Affiliations:** Department of Psychology, College of Education and Sports Sciences, Yangtze University, Jingzhou 434023, China; 2024710083@yangtzeu.edu.cn (S.Z.); 2024710078@yangtzeu.edu.cn (J.W.);

**Keywords:** dark triad, bullying victimization, cyberbullying, Giants on the Internet

## Abstract

A significant body of research has documented the aggressive and antisocial tendencies of individuals with dark triad personality traits. Although the prevalence of dark personalities in online environments is often criticized, there is a need to explore effective strategies to mitigate or stop such behaviors. This study aims to shed light on the intriguing phenomenon of “Giants on the Internet, cowards in real life” by examining the longitudinal relationship between dark triad traits, bullying victimization, and cyberbullying. Study 1 revealed that adolescents tend to display heightened tendencies towards cyberbullying after experiencing real-life victimization. Study 2, on the other hand, showed a reduction in cyberbullying behaviors among those with dark triad traits following experiences of bullying. These findings highlight the paradoxical mechanisms underlying the relationship between bullying victimization, dark triad traits, and cyberbullying. Consequently, this study introduces the new label, “From Online Aggression to Offline Silence,” to describe this dynamic.

## 1. Introduction

With the advancement of technology, traditional school bullying has transcended the school setting into cyberspace for decades ([47]; [7]). Cyberbullying refers to intentional and repeated harm inflicted through electronic communication technologies, such as social media or messaging platforms. Prior research consistently demonstrates that cyberbullying victimization can significantly undermine adolescents’ psychological, physical, and behavioral well-being ([30]). Victims often experience anxiety, depression, and social withdrawal, while perpetrators may derive pleasure or a sense of control from their aggressive acts ([43]; [8]). Therefore, understanding the antecedents and psychological mechanisms that underlie cyberbullying perpetration is crucial for designing effective prevention strategies.

Among the most relevant personality frameworks explaining aggressive behaviors is the Dark Triad—a constellation of three socially aversive traits: Machiavellianism, psychopathy, and narcissism ([18]). Narcissism is characterized by grandiosity, entitlement, and a strong need for admiration; Machiavellianism reflects manipulativeness, strategic exploitation, and emotional detachment; and psychopathy denotes impulsivity, callousness, and a lack of empathy or remorse. Collectively, these traits are positively related to antisocial, exploitative, and aggressive behaviors, including both offline and online aggression ([9]; [11]; [42]). Empirical evidence further supports these links: individuals high in Dark Triad traits are more prone to engage in criminal or norm-violating acts ([13]) and exhibit reduced altruistic tendencies and environmental concern ([33]; [10]). Conversely, these traits are negatively associated with prosocial and empathic behaviors, as individuals high in these characteristics often lack concern for others and for broader social or ecological systems.

While substantial research has documented the direct relationship between Dark Triad traits and cyberbullying, less is known about how contextual factors—such as experiences of victimization—may shape or moderate this association. Theoretical and empirical studies suggest that individuals exposed to external stressors or real-life victimization often adjust or withdraw from certain behaviors as coping responses ([38]). Drawing on this perspective, the present study proposes that real-life bullying victimization may moderate the relationship between Dark Triad traits and cyberbullying perpetration. Specifically, it is proposed that adolescents with high levels of Dark Triad traits may be less likely to engage in online aggression after experiencing offline victimization—a dynamic encapsulated in the phrase “From Online Aggression to Offline Silence.” To better understand this mechanism, the current research aims to examine the longitudinal moderating effects of bullying victimization on the association between Dark Triad traits and cyberbullying behaviors among adolescents.

### 1.1. From Victimization to Cyberbullying

Bullying victimization refers to an adolescent experiencing continual physical, verbal and relational hurts by peers in school ([34]). Numerous studies have determined that real-world bullying can result in behavioral and mental health issues. For instance, researchers have found that students experiencing bullying reported higher levels of depression, anxiety and stress ([23]; [41]). Existing research has confirmed two extreme behavioral tendencies: withdrawal and aggression, with aggressive behaviors being linked to victimization ([32]). Moreover, another study has found that cybervictimization was strongly related to cyberbullying ([1]). 

**Hypothesis** **1:**
*There is a longitudinal relationship between real-life victimization and cyberbullying.*


Specifically, experiences of real-life victimization may predict an increase in subsequent cyberbullying behaviors.

### 1.2. The Potential Link Between Dark Triad and Cyberbullying

With the widespread use of social media platforms and smartphones, cyberbullying has become increasingly severe, causing significant harm to adolescents. Cyberbullying typically refers to deliberate and repetitive harm inflicted by one or more peers, occurring in cyberspace through the use of computers, smartphones, and other devices ([57]; [25]). Recently, various subtypes of cyberbullying, including doxxing, impersonation, and cyberstalking, have emerged ([56]; [54]; [36]). Research shows that cyberbullying participation may stem from various factors such as the need for social acceptance, envy, and personality differences ([21]). The Dark Triad consists of three distinct but related personality traits, which involve manipulative behavior, lack of empathy, and a desire for power or admiration, traits that may contribute to engaging in cyberbullying ([27]; [17]). These traits are also linked to potential peer conflicts and feelings of alienation ([40]). Additionally, research discovered a substantial correlation between cyberbullies and a high propensity for violence ([50]; [45]), a lack of empathy ([59]), and poor self-control ([4]). Consequently, personality characteristics would be crucial in explaining this violent conduct, with dark personalities emerging as the most significant predictor of it ([51]; [13]). 

**Hypothesis** **2:**
*The dark triad can substantially strengthen cyberbullying over time.*


Specifically, individuals with higher levels of Machiavellianism, narcissism, or psychopathy are more likely to engage in future cyberbullying behaviors.

### 1.3. The Moderating Role of Real-Life Victimization

Preliminary research indicates a strong link between dark triad traits and both bullying and cyberbullying ([11]). In response, various societal sectors have sought to curb the rise in cyberbullying by implementing strategies informed by contemporary research ([15]). However, despite these efforts, cyberbullying remains a significant threat to the health of the online environment ([53]). While many studies have focused on positive interventions to alleviate and improve this situation, there has been limited exploration of alternative perspectives. In China, certain widely used idioms offer intriguing insights that may inspire new approaches to addressing this issue. Phrases such as “以暴制暴” (fight violence with violence) and “以毒攻毒” (use poison to fight poison) suggest unconventional methods for dealing with negative behaviors. These expressions reflect a philosophy that sometimes, countering aggression with similarly forceful measures can be effective.

**Hypothesis** **3:**
*Experiencing high level of bullying victimization in reality may moderate the relationship between dark triad traits and cyberbullying.*


### 1.4. The Current Study

The current study comprised two sub-studies designed to explore the longitudinal relationships among bullying victimization, Dark Triad traits, and cyberbullying. Based on the existing theories and research, a cross-lagged panel model (CLPM) was used to investigate the longitudinal relation between victimization and cyberbullying in study 1. Study 2 focused on the influence of dark triad personality traits on cyberbullying, as well as the moderating role of victimization in this relationship over time. The findings confirmed the hypothesis that individuals with dark triad traits are “Giants on the Internet, cowards in real life,” offering new insights into the mechanisms that drive cyberbullying behavior.

## 2. Methods

### 2.1. Participants and Procedures

To facilitate longitudinal data tracking, several public middle schools near the research institution were invited to participate in this study. Students’ grade levels and classes were randomly selected, and a number of students from each selected class were invited to participate. Data were collected using a paper-and-pencil questionnaire. With the assistance of the head teachers, trained research assistants guided the participants in completing the surveys in classroom settings. Both students and their parents provided informed consent prior to participation. The final sample consisted of 606 students (343 boys and 263 girls), aged 12–17 years (M = 14.88, SD = 1.68). The first wave of data collection took place in November 2022, and the second wave was conducted in January 2024. After two waves of investigation, 596 students remained in the longitudinal sample. This study received ethical approval from the Ethics Committee of the Department of Psychology at the authors’ institution.

### 2.2. Measures

#### 2.2.1. Bullying Victimization (BV)

To measure the experiences of bullying victimization, we used the traditional bullying scale developed by [34] ([34]). The scale consists of two subscales: the scale of Bullying and the scale of Victimization. The sub-scale of bullying victimization was translated into a Chinese version and mainly used to assess the level of victimization of adolescents. This scale contains 6 items, and each item is scored on a 6-point Likert scale ranging from 1 (never) to 6 (always). Higher scores indicated the intensive frequency of BV. The translated version has been proven with good internal consistency by a large number of studies in China, and the Cronbach’s alpha for the whole scale was 0.863.

#### 2.2.2. Dark Triad (DT)

The Dark Triad Scale (DTS), as developed by the Dirty Dozen ([17]; [19]), was utilized in this study to assess hateful personality traits. This scale consists of 12 items, divided into three dimensions representing the dark triad. Each item is rated on a 7-point Likert scale, ranging from 1 (totally disagree) to 7 (fully agree). The DTS was translated into Chinese and demonstrated good consistency and validity in previous research ([39]). Higher scores reflect high levels of DT, and the Cronbach’s alpha for the whole scale was 0.899.

#### 2.2.3. Cyberbullying (CB)

Cyberbullying was assessed by the E-Bullying Scale (E-BS) developed by [24] ([24]). It has 6 items to measure the extent of cyberbullying; each participant reported their cyberbullying behaviors from 0 (never) to 6 (more than six times). To better conduct the data collection process, the original scale was translated into the Chinese version, and previous research has established its good reliability and validity ([15]). The present study’s reliability coefficient for the CB was 0.902.

#### 2.2.4. Demographic Variables

Respondents’ gender was measured by asking whether they were boys or girls (1 = boy, 2 = girl). Also, all participants reported their grade numbers from grade 1 to grade 3. Moreover, age was included at T1, along with some other basic information.

#### 2.2.5. Common Method Bias

All the data were collected by surveys and questionnaires, which may lead to a common method bias. According to Harman’s factor analysis method, the first factor explains 20.83% of the variance, which does not exceed the commonly accepted threshold of 40%. Thus, it can be tentatively concluded that this study does not exhibit significant common method bias.

### 2.3. Analytic Strategy

Statistical analyses were conducted using IBM SPSS Statistics (Version 26.0; IBM Corp., Armonk, NY, USA) and Mplus 8.3 from Muthén & Muthén. We initially performed descriptive statistics of the main study variables and then performed a correlation analysis to capture the variance across different time points. To examine the paradoxical phenomenon about “From Online Aggression to Offline Silence”, the longitudinal relation between BV, DT and CB requires further investigation. Study 1 aims to build a pathway model to reveal the longitudinal development from BV to CB. Study 2 mainly focuses on the specific group of individuals with dark triad personality, the relation between DT and CB, and the moderating effect of BV.

## 3. Results

### 3.1. Preliminary Analyses

Descriptive statistics and bivariate correlations among the main variables across two time points are presented in Table 1. All variables were positively correlated with each other across time points (r = 0.09–0.47, *p* < 0.01), indicating moderate to strong associations among bullying victimization, Dark Triad traits, and cyberbullying. To conduct the invariance measurement, this study calculated the configural, metric and scalar invariance across gender in Table 2. The results showed the main variables differed slightly across gender, but no significant differences were found.

### 3.2. Study 1: From Victimizations to Cyberbullying

The CLPM (M1) in study 1 had an acceptable fit (χ2 = 2.391, CFI = 0.997, RMSEA = 0.047, SRMR = 0.015), and the path coefficients are presented in Table 3. At the within-person level, the autoregressive process on each variable indicated almost strong associations between T1 and T2. At the between-person level, BV was positively associated with CB (r > 0, *p* < 0.01) over time. Specifically, BV at T1 had a positive effect on CB at T2 (β = 0.200, *p* < 0.001). Similarly, BV at T2 was positively influenced by CB at T1 (β = 0.253, *p* < 0.001). This result indicated that the cross-lagged effects were significant and Hypothesis 1 was supported.

It is noteworthy to mention that study 1 documented the longitudinal progress from BV to CB (as seen in Figure 1). This finding suggested that individuals might report a higher inclination to CB after victimization in real life. And this aggressive behavior on the Internet was time-sensitive; the closer it occurs to the time of victimization, the stronger the association between experiencing BV in reality and engaging in CB. Conversely, this correlation gradually weakens as time passes.

### 3.3. Study 2: The Moderating Effects of Victimizations

To achieve a more profound comprehension of dark personality dynamics in the context of cyberbullying. Study 2 constructed a longitudinal moderating model to examine the relation between DT and CB and the moderating effect of BV (Figure 2). The CLPM (M2) in study 2 also fit the data well (χ2 = 37.524, RMSEA = 0.094, CFI = 0.954, SRMR = 0.032), and the coefficients of the paths are exhibited in Table 4. At the within-person level, autoregressive processes on DT (b_t1–t2_ = 0.435, *p* < 0.05) and CB (b_t1–t2_ = 0.472, *p* < 0.05) showed a strong correlation between T1 and T2. At the longitudinal level, DT had a positive effect on CB (r > 0, *p* < 0.05) across time points, which supported Hypothesis 2. More specifically, DT at T1 had a positive effect on CB at T2 (β = 0.080, *p* < 0.05). However, CB had no significant effect at T1 (β = 0.018, *p* > 0.05).

As shown in Table 4, the interaction effect of DT (T1) and BV (T1) was significant (β = −0.086, *p* < 0.05), and BV at T2 also played a moderating role (β = −0.080, *p* < 0.05) in the longitudinal relation between DT (T1) and CB (T2). Consequently, the simple slope test needed to be performed to gain more information, as shown in Figure 3.

The results of the interaction effect indicated that the association between DT (T1) and CB (T2) was weaker when individuals experienced higher levels of BV in reality. This finding suggested that individuals with dark personality traits would decrease their cyberbullying behaviors on the Internet after being victims in real life. However, those with a lower level of the Dark Triad exhibited a higher inclination toward cyberbullying behaviors after being bullied in reality. In this regard, the findings support the Hypothesis 3 in this research.

### 3.4. Additional Findings

With the exception of findings from two sub-studies, the longitudinal correlation between BV and DT remained statistically significant, as shown in Table 1. This indicated that adolescents who usually experience BV might report higher levels of DT. Moreover, the longitudinal relation between BV and DT and the potential mediation effect of DT between BV and CB were also confirmed. As seen in Table 5, BV (T1) was positively related to DT (T2), and DT (T2) had a positive effect on BV (T1). In addition, DT (T1&T2) might play the mediation role in the longitudinal relation between BV (T1) and CB (T2). The results of mediation analysis indicated significant effects (b = 0.030, SE = 0.012, 95% CI = [0.013, 0.053]).

## 4. Discussion

This study explored the longitudinal relationships between bullying victimization, dark triad traits, and cyberbullying among adolescents, revealing both mediating and moderating mechanisms. The findings suggest that adolescents who experience victimization are more likely to express their anger through cyberbullying rather than seeking direct revenge in real life. Additionally, the moderating effect of victimization showed a weakening of the relationship between dark triad traits and cyberbullying, particularly in adolescents with pronounced dark personality traits. This supports the identified phenomenon of “Giants on the Internet, cowards in real life.”

### 4.1. Online Actions over Real-Life Confrontation

As information and technology have advanced, a multitude of social media platforms have emerged, offering users incredible convenience in accomplishing their desired goals ([20]). However, recent research has shown that the anonymity of the internet can lead to a loss of self-control in users’ behavior ([6]). For instance, studies suggest that individuals are more likely to engage in uncivil or aggressive actions when shielded by the anonymity of the online environment ([29]; [44]). Adolescents, in particular those born into the era of social media, often struggle to navigate the boundaries between appropriate and inappropriate behavior during this formative stage of life ([52]). Furthermore, the consequences of online behaviors are perceived to be far less severe than those of offline actions, as real-world outcomes are often irreversible, leading to significant impacts on both parties involved ([26]; [22]). Conversely, the immediacy and reversibility of online actions lower the costs associated with undesirable behaviors, allowing teenagers to promptly adjust or retract what they did according to their own will. This makes it easier for them to pursue their goals in the online space. When the internet can no longer shield them, however, they may lose direction and revert to being “nobodies” in the real world.

### 4.2. The Process from Bullying Victimization to Cyberbullying

Study 1 identified a longitudinal relationship between real-life victimization and cyberbullying and also highlighted the potential mediating role of Dark Triad traits. Previous research has shown that victimization in online environments is closely linked to cyberbullying, often mediated by feelings of anger ([1]). This mechanism similarly explains the pathway from bullying victimization (BV) to cyberbullying (CB) in the current study. Furthermore, adolescents often perceive bullying victimization as a hostile threat ([48]), prompting them to seek help from peers ([14]). However, research suggests that victims frequently struggle to receive support, especially in healthy peer contexts, as peers may be unaware of the truth or how to provide assistance ([41]; [35]). Suffering the associated pains of bullying and social exclusion hurts them too much, and they are more inclined to become indignant and defend themselves ([5]). Lacking real-world support, they turn to cyberspace as a refuge to vent their frustration and resentment. Additionally, studies have shown that individuals derive significant satisfaction from engaging in aggressive behaviors ([43]), which is closely related to the concept of psychological compensation ([31]). Therefore, cyberbullying becomes the preferred outlet for releasing frustrations once victimization occurs ([55]). Another finding from study 1 suggests that dark triad traits may act as a mediator between cyberbullying and victimization, offering a new perspective on this process. Evidence indicates that certain life events, especially negative ones, can contribute to shifts in personality traits ([3]). Furthermore, a high correlation was found between bullying incidents and the dark triad ([11]). This suggests that the dark personality may be activated by experiencing victimization, leading teenagers to seek revenge through cyberbullying.

### 4.3. Dark Triad and Cyberbullying: The Paradox of Victimization

Study 2 further explored the paradoxical phenomenon by examining the longitudinal impact of victimization on the relationship between dark triad traits and cyberbullying. It was found that individuals with dark personality traits are strongly associated with cyberbullying ([11]). However, the moderating effect of victimization, as observed in study 2, revealed a decrease in cyberbullying behaviors among those with dark triad traits. This suggests that stress and perceived threat can lead to reduced engagement in both social and personal domains ([37]). According to cognitive-behavioral theory, an individual’s behavior is significantly shaped by their cognitive evaluations of situations before actions occur ([16]). Recent research also suggests that risk appraisals can alter one’s intentions and subsequent behaviors ([46]). For adolescents, real-life victimization, as a major stressful life event, forces them to deeply reflect on the potential outcomes and consequences of engaging in cyberbullying ([48]). Moreover, studies have shown that cyberbullying is often driven by recreational motives ([12]). Thus, adolescents are pragmatic enough to avoid greater real-world conflicts, embodying the proverb “Giants on the Internet, cowards in real life.”

Although previous theories, such as the frustration–aggression hypothesis, posit that individuals experiencing real-life frustration are more likely to express aggression, this assumption may not fully apply to the context of online behavior ([58]). Those who suffer significant setbacks in reality often lack the perceived power or psychological safety to express hostility, even in virtual spaces. Instead, they tend to remain silent or withdraw from online interactions due to fear of social judgment or retaliation ([2]). In contrast, the so-called “online giants” who engage in aggressive or dominating behaviors on the Internet are typically characterized by a relatively strong sense of agency and social participation. Their behavior may not stem from actual powerlessness but rather from a perceived threat to social status or control. From the perspective of compensatory self-enhancement and power motivation theories, such individuals may use the online environment as a safe arena to restore their sense of dominance and self-worth by imposing control or moral pressure on others. This distinction highlights that online aggression may reflect not the helplessness of the defeated but the compensatory assertion of those experiencing relative deprivation or threatened superiority.

### 4.4. Implications

This study reveals a compelling paradox in adolescent behavior: individuals exhibiting pronounced Dark Triad traits—typically associated with heightened aggression and online hostility—tend to reduce their cyberbullying behaviors following experiences of personal victimization. This inverse relationship highlights the intricate interplay between aggression and vulnerability, supporting the notion that online aggressors may exhibit fragility when confronted with real-life adversity.

From a theoretical standpoint, these findings advance the understanding of the self-regulatory and emotional mechanisms underlying cyber aggression. Experiencing victimization may trigger empathy, fear of retaliation, or self-protective withdrawal, consistent with social learning theory and emotion regulation perspectives. Such processes may attenuate the expression of antisocial tendencies, suggesting that exposure to real-world consequences can recalibrate behavioral impulses shaped by Dark Triad traits.

Practically, these results underscore the need for interventions that address the psychological roots of cyber aggression—particularly emotional distress, perceived injustice, and the desire for revenge. Programs fostering emotional resilience, empathy training, and reflective awareness of one’s online behavior could prove effective in curbing cyberbullying. More broadly, this work contributes to a nuanced understanding of how personality, social context, and lived experiences interact to shape aggression across digital and physical domains.

## 5. Limitations and Future Directions

There are several limitations in the current study that should be considered. Firstly, the study only followed two waves of data, which may limit the robustness of the conclusions. Secondly, while the study highlighted an intriguing online phenomenon, it centered exclusively on the issue of cyberbullying. In fact, there are other aspects that could similarly illustrate this phenomenon but were not explored in this study. For instance, future investigations could focus on the discrepancies between online and offline self-disclosure behaviors to further demonstrate this interesting phenomenon. Furthermore, existing research suggests that while the Dark Triad traits overlap to some extent, they remain distinct, with each trait potentially leading to different outcomes ([28]; [49]; [27]). Therefore, future research could benefit from a deeper exploration of how each of the Dark Triad traits manifests specifically in bullying contexts, helping to clarify their unique roles and impacts.

## 6. Conclusions

To conclude, the present study proposes a new perspective on understanding the interplay between Bullying Victimization (BV), Dark Triad (DT) traits, and Cyberbullying (CB). Study 1 demonstrated the longitudinal development from BV (T1) to CB (T2) over time (supporting Hypothesis 1), and study 2 provided evidence on the effect of dark personality on cyberbullying (supporting Hypothesis 2) and the moderating role of BV in the longitudinal relationship between DT (T1) and CB (T2) (supporting Hypothesis 3). The findings suggest that adolescents are more likely to engage in cyberbullying rather than directly seeking revenge in person.

## Figures and Tables

**Figure 1 behavsci-15-01583-f001:**
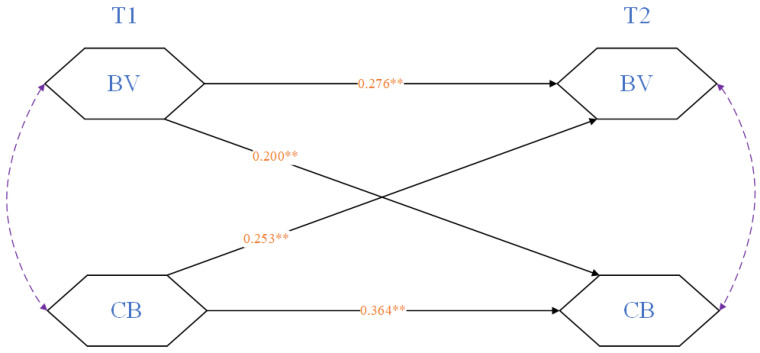
CLPM (M1): The longitudinal progress from BV to CB. ** = *p* < 0.01. Same as below.

**Figure 2 behavsci-15-01583-f002:**
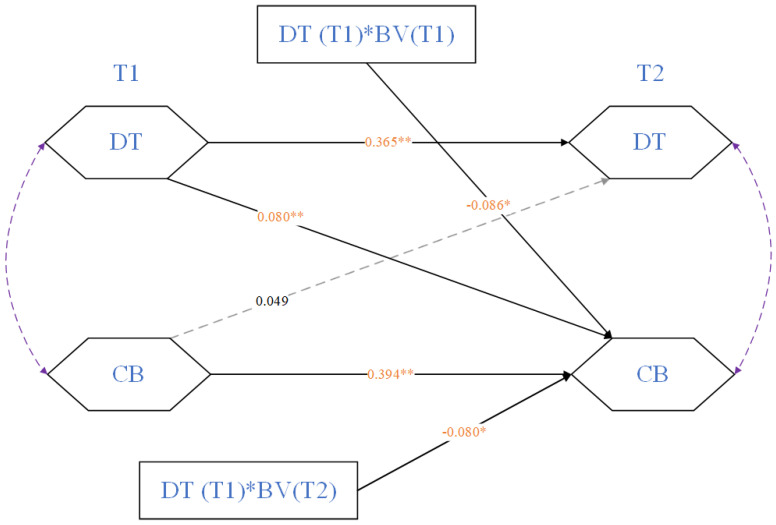
CLPM (M2): The longitudinal relation between DT and CB and the moderating effect of BV. * = *p* < 0.05, ** = *p* < 0.01. Same as below.

**Figure 3 behavsci-15-01583-f003:**
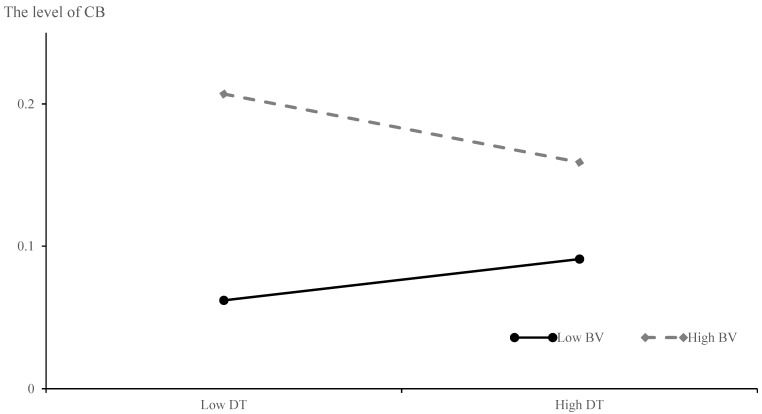
Moderating Effect of BV on the Longitudinal Relationship Between DT (T1) and CB (T2).

**Table 1 behavsci-15-01583-t001:** Means, standard deviations, and correlation of main variables.

	T1 BV	T2 BV	T1 DT	T2 DT	T1 CB	T2 CB
T1 BV	-					
T2 BV	0.394 **	-				
T1 DT	0.215 **	0.146 **	-			
T2 DT	0.134 **	0.115 **	0.363 **	-		
T1 CB	0.467 **	0.382 **	0.218 **	0.095 *	-	
T2 CB	0.216 **	0.397 **	0.149 *	0.198 **	0.385 **	-
M	1.206	1.223	2.481	2.553	0.087	0.136
SD	0.385	0.433	0.997	1.189	0.425	0.507

**Note.** BV = Bullying Victimization, DT = Dark Triad, CB = Cyberbullying, T1 = at time 1, T2 = at time 2, * = *p* < 0.05, ** = *p* < 0.01. same as below.

**Table 2 behavsci-15-01583-t002:** Measurement invariance and model fit circumstances.

Model	*χ2*	*CFI*	Δ*CFI*	*RMSEA*	*SRMR*	Δ*RMSEA*
configural	6.314	0.989		0.064	0.027	
metric	7.555	0.985	−0.004	0.055	0.032	−0.009
scalar	10.074	0.983	−0.002	0.048	0.046	−0.007
M1	2.391	0.997		0.047	0.015	
M2	37.524	0.954		0.094	0.032	

**Note.** χ2 = Chi-Square, CFI = Comparative Fit, RMSEA = Root Mean Square Error of Approximation, SRMR = Standardized Root Mean Square, Δ = difference.

**Table 3 behavsci-15-01583-t003:** CLPM: Cross-lagged effects of BV and CB.

	T2 BV				T2 CB			
	*b*	*SE*	*β*	*p*	*b*	*SE*	*β*	*p*
T1 BV	0.311	0.046	**0.276**	<0.001	0.264	0.053	**0.200**	<0.001
T1 CB	0.257	0.042	**0.253**	<0.001	0.433	0.051	**0.364**	<0.001

**Note.** BV = Bullying Victimization, CB = Cyberbullying, significant coefficient is in bold.

**Table 4 behavsci-15-01583-t004:** CLPM: Cross-lagged effects of DT and CB, and the moderating effect of BV.

	T2 DT				T2 CB			
	*b*	*SE*	*β*	*p*	*b*	*SE*	*β*	*p*
T1 DT	0.435	0.045	**0.365**	<0.001	0.067	0.019	**0.080**	0.031
T1 CB	0.049	0.109	0.018	0.653	0.472	0.049	**0.394**	<0.001
T1 DT × T1 BV					−0.100	0.047	**−0.086**	0.033
T1 DT × T2 BV					−0.074	0.037	**−0.080**	0.041

**Note.** DT = Dark Triad, CB = Cyberbullying, BV = Bullying Victimization, significant coefficients are in bold.

**Table 5 behavsci-15-01583-t005:** CLPM: Cross-lagged effects of BV and DT, and the mediation effect of DT.

	T2 DT			T2 BV			T2 CB		
	*b*	*SE*	*p*	*b*	*SE*	*p*	*b*	*SE*	*p*
T1 BV	**0.131**	0.040	0.001	**0.394**	0.035	<0.001	**0.176**	0.040	<0.001
T1 DT	**0.337**	0.035	<0.001	0.046	0.036	0.209	0.058	0.043	0.176
T2 DT							**0.127**	0.039	<0.001
	*χ2*			*CFI*		*RMSEA*	*SRMR*		
Model Fit		7.447		0.996		0.029		0.023	

**Note.** DT = Dark Triad, CB = Cyberbullying, BV = Bullying Victimization, significant coefficient is in bold.

## Data Availability

The data that support the findings of this study are available from the corresponding author upon reasonable request.

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
