# Peer review of "From Online Aggression to Offline Silence: A Longitudinal Examination of Bullying Victimization, Dark Triad Traits, and Cyberbullying"

_behavsci, 2025, doi:10.3390/bs15111583_

Round 1
Reviewer 1 Report
Comments and Suggestions for Authors
The manuscript addresses an important and timely topic. Specifically, across two studies, the authors investigated the longitudinal relation between victimisation and cyberbullying (Study 1) as well as the influence of dark triad personality traits on cyberbullying, and the moderating role of victimisation in this relationship over time (Study 29. Despite its potential, this article has several limitations that undermine its publication.
Regarding the introduction, I agree with a warm-up on the current situation regarding technology advancement. However, the introduction should clearly explain the relationships and mechanisms addressed in this study from a theoretical perspective. Therefore, the authors should provide: a section on the relationship between victimisation and cyberbullying, a section on the relationship between the Dark Triad and cyberbullying, and a section on the moderating role of victimisation. In addition, the authors should provide more detailed definitions of the constructs addressed in their work. For instance, the description of the Dark Triad appears to be drafted. Please provide the main features for each trait and their positive association with maladaptive behaviours, as well as negative relationships with practices aimed at taking care of individuals and their environment. Please include and describe these references:
Hampejs, V., Zwickl, A. A., Tran, U. S., & Voracek, M. (2025). The Dark Triad of personality and criminal and delinquent behavior: Preregistered systematic review and three-level meta-analysis. Personality and Individual Differences, 246, 113308.
Oda, R., & Matsumoto-Oda, A. (2022). HEXACO, Dark Triad and altruism in daily life. Personality and Individual Differences, 185, 111303.
Giancola, M., Palmiero, M., & D’Amico, S. (2023). The association between Dark Triad and pro-environmental behaviours: the moderating role of trait emotional intelligence (La asociación entre la Tríada Oscura y las conductas proambientales: el papel moderador de la inteligencia emocional rasgo). PsyEcology, 14(3), 338-362.
For each variable addressed in this research, please use the same logic.
Moreover, I suggest providing a more detailed literature review.
The present study aims to inform about the logic of the research, providing a precise formulation of the research hypothesis and stressing the theoretical framing. In addition, please drop "The findings confirmed the hypothesis that 98 individuals with dark triad traits are 'Giants on the Internet, cowards in real life,' offering new insights into the mechanisms that drive cyberbullying behaviour". Please
Please do not use questions as headings.
Please provide more details about data gathering to ensure transparency.
For each measure used, please provide psychometric properties.
In the discussion, provide a more detailed explanation of the implications of this research, focusing on both theoretical and practical contributions.
Author Response
Comments 1:
Regarding the introduction, I agree with a warm-up on the current situation regarding technology
advancement. However, the introduction should clearly explain the relationships and
mechanisms addressed in this study from a theoretical perspective. Therefore, the authors
should provide: a section on the relationship between victimisation and cyberbullying, a section
on the relationship between the Dark Triad and cyberbullying, and a section on the moderating
role of victimisation. In addition, the authors should provide more detailed definitions of the
constructs addressed in their work. For instance, the description of the Dark Triad appears to be
drafted. Please provide the main features for each trait and their positive association with
maladaptive behaviours, as well as negative relationships with practices aimed at taking care of
individuals and their environment. Please include and describe these references:
Hampejs, V., Zwickl, A. A., Tran, U. S., & Voracek, M. (2025). The Dark Triad of personality and
criminal and delinquent behavior: Preregistered systematic review and three-level meta-analysis.
Personality and Individual Differences, 246, 113308.
Oda, R., & Matsumoto-Oda, A. (2022). HEXACO, Dark Triad and altruism in daily life. Personality
and Individual Differences, 185, 111303.
Giancola, M., Palmiero, M., & D’Amico, S. (2023). The association between Dark Triad and
pro-environmental behaviours: the moderating role of trait emotional intelligence (La asociación
entre la Tríada Oscura y las conductas proambientales: el papel moderador de la inteligencia
emocional rasgo). PsyEcology, 14(3), 338-362.
Response 1:We sincerely appreciate the reviewer’s constructive comments. Incorporating the suggested revisions has substantially improved the theoretical clarity and coherence of the Introduction. The enhanced conceptual definitions and theoretical discussions now provide a stronger foundation for understanding the mechanisms explored in this study, thereby increasing the overall rigor and readability of the manuscript.
Comments2: For each variable addressed in this research, please use the same logic. Moreover, I suggest providing a more detailed literature review
Response2:We appreciate the reviewer’s helpful suggestion. Following this advice has improved the internal consistency and depth of the literature review. By applying a coherent logic across all variables and expanding the relevant literature, the theoretical rationale of the study has become more systematic and comprehensive.
Comments3:The present study aims to inform about the logic of the research, providing a precise formulation of the research hypothesis and stressing the theoretical framing. In addition, please drop "The findings confirmed the hypothesis that 98 individuals with dark triad traits are 'Giants on the
Internet, cowards in real life,' offering new insights into the mechanisms that drive cyberbullying
behaviour".
Response3:We sincerely thank the reviewer for this valuable suggestion. The revision has helped us present the research logic and hypotheses with greater precision and emphasize the theoretical framing more effectively. Removing the mentioned sentence also enhanced the academic tone and objectivity of the manuscript.
Comments4:Please do not use questions as headings.
Response4:We appreciate the reviewer’s helpful reminder. The headings have been revised to declarative statements, which improves the manuscript’s academic tone and structural clarity.
Comments5:Please provide more details about data gathering to ensure transparency
Response5:We appreciate the reviewer’s thoughtful comment. We have provided more detailed information about the data collection process to enhance transparency and replicability.
Comments6:For each measure used, please provide psychometric properties.
Response6:We appreciate the reviewer’s valuable suggestion. We have added detailed descriptions of the psychometric properties (e.g., reliability and validity) for each measure used in this study, which improves the methodological transparency and strengthens the credibility of our findings.
Comments7:In the discussion, provide a more detailed explanation of the implications of this research, focusing on both theoretical and practical contributions.
Response7:Thank you for pointing this out. We agree with that comment. We add theoretical and practical significance to the article, especially the practical significance of intervention. Lines 327-347.
Reviewer 2 Report
Comments and Suggestions for Authors
The first sentence seems to indicate cyberbullying is a new behavior. Please clarify that it has been in existence for decades.
Also, be sure to look through the document carefully in relation to grammatical errors and lack of clarity.
Line 48 has the word "things" within a sentence, which is unnecessary and grammatically incorrect.
The words, "existed researches" are used instead of "existing researchers have" confirmed. This needs to be corrected. See line 50.
That same sentence (50) is very unclear, as it is difficult to interpret what the difference is between "aggressions" and
aggressive behavior."
Again, please take time to go through the entire document, looking for similar errors.
Given the fact that cyberbullying is one of the main concepts, it is important to dig deeper into the definition and the various subtypes. It's far more than what you have indicated. Additional examples are needed (e.g., doxxing, impersonation, ignoring, etc.).
Author Response
Comments 1: The first sentence seems to indicate cyberbullying is a new behavior. Please clarify that it has been in existence for decades.
Response 1: Thank you for pointing this out. We agree with this comment. Therefore, we have revised the first sentence to "With the advancement of technology, traditional school bullying has transcended the school setting into cyberspace for decades" to clarify that cyberbullying has been in existence for decades.
Comments 2: Also, be sure to look through the document carefully in relation to grammatical errors and lack of clarity.
Response 2: Agree. We have changed "level" to "levels" in line 49, as it refers to the varying levels of three distinct conditions: depression, anxiety, and stress.
Comments 3: Line 48 has the word "things" within a sentence, which is unnecessary and grammatically incorrect.
Response 3: Thank you for pointing this out. We agree with this comment. We have removed "the" and "things" and used the phrase "experiencing bullying" directly.
Comments 4: The words, "existed researches" are used instead of "existing researchers have" confirmed. This needs to be corrected. See line 50.
Response 4: Thank you for pointing this out. We agree with this comment. Therefore, we have changed “existed” to “existing”.
Comments 5: That same sentence (50) is very unclear, as it is difficult to interpret what the difference is between "aggressions" and aggressive behavior."
Response 5: Thank you for your valuable feedback. In response to your comment regarding the terms 'aggressions' and 'aggressive behaviors,' We have revised the sentence to clarify that there is no significant distinction between the two in the context of this study.
The specific modifications are as follows:
”The existing researches have confirmed two extreme behavioral tendencies: withdrawal and aggression, with aggressive behaviors being linked to victimization.”
Comments 6: Given the fact that cyberbullying is one of the main concepts, it is important to dig deeper into the definition and the various subtypes. It's far more than what you have indicated. Additional examples are needed (e.g., doxxing, impersonation, ignoring, etc.).
Response 6: Thank you for pointing this out. We agree with this comment. Therefore, We have provided an in-depth explanation of the definition of cyberbullying and discussed some of its subtypes. Specific modifications are as follows:
With the widespread use of social media platforms and smartphones, cyberbullying has become increasingly severe, causing significant harm to adolescents. Cyberbullying typically refers to deliberate and repetitive harm inflicted by one or more peers, occurring in cyberspace through the use of computers, smartphones, and other devices (Zhu et al., 2021; Leduc et al., 2022). Recently, various subtypes of cyberbullying, including doxxing, impersonation, cyberstalking, etc., have emerged (Zhou et al., 2024; Vranda et al., 2023; Pereira & Matos, 2016).

Reviewer 3 Report
Comments and Suggestions for Authors
The subject is relevant and worthy of attention. The references are updated and relevant. However, there are a few points which need to be revised and, eventually, expanded.
Even though the subject is adequately explained and the aims are clearly detailed, the hypotheses (H1 and H2) are not as clearly stated and formulated. It would also benefit the study if they were retrieved in the conclusion, stating whether the data collected and its discussion prove or disprove them.
More detailed comments:
- I was unclear on how the quote ‘Giants on the Internet, cowards in real life’ should be understood. Is this a saying? Did the authors coin it for the purposes of this article? Is it one of the hypotheses (as the text in 1.4. seems to indicate)? This should be made clear, especially because the expression is repeated so often throughout the text.
- 48: ‘the bullying things…’. Can you be more specific?
- 50: I believe it should be ‘existing’ (not ‘existed’)
- 52: ‘being cybervictimisation…’ Can you please rephrase?
- 53 and 54: linguistic corrections are necessary
- In 1.3., I did not find these surprising, nor did I find them something that can be described as specifically Chinese. There are similar idioms in other languages, which makes me think this might be more of a generalised perspective
- I have some doubts as to the methodology and data collection. I believe the manuscript would benefit from some expansion, particularly in the reasons for selecting these age cohorts. Justifying this at this point would make it easier to properly identify the study's limitations and point out possibilities for further research at a later stage. What were the criteria for inclusion and exclusion? What were the questions asked? How were the questionnaires applied?
- Still on the same issue, why were there two phases (‘waves’?) for this study? Why was there such a long gap between them? Could you please expand on this? And why did 596 remain? Why were the others excluded?
- In 2.2.1., the date reference for the Olweus scale is missing here
- 141: lexical inconsistency: ‘contained’?
- The conclusion is relatively short. I believe it could be improved by either expanding it or incorporating in it the limitations and suggestions for further studies. If that were the case, the conclusion section could begin with this part, be followed by the limitations and, afterwards, would come the suggestions for further studies. This structure would enhance a sense of progression in the argumentation. As it stands, the conclusion sounds somewhat repetitive.
The manuscript requires a very thorough linguistic revision. There are syntactical and punctuation issues which sometimes hinder understanding. There are also some typos which should be addressed.
Author Response
Comments 1: Even though the subject is adequately explained and the aims are clearly detailed, the hypotheses (H1 and H2) are not as clearly stated and formulated. It would also benefit the study if they were retrieved in the conclusion, stating whether the data collected and its discussion prove or disprove them.
Response 1:We appreciate the reviewer’s valuable comment. The hypotheses (H1 and H2) have been clearly reformulated and explicitly stated in the revised manuscript. In addition, they are revisited in the conclusion section to indicate whether the results support or refute them, enhancing the coherence between the study’s aims, findings, and conclusions.
Comments 2:I was unclear on how the quote ‘Giants on the Internet, cowards in real life’ should be understood. Is this a saying? Did the authors coin it for the purposes of this article? Is it one of the hypotheses (as the text in 1.4. seems to indicate)? This should be made clear, especially because the expression is repeated so often throughout the text.
Response 2:Thank you for pointing this out. We agree with that comment. The phrase was originally coined by Lenin, referring to individuals who appear ambitious and eloquent in thought but lack corresponding action or determination. In this study, the expression “Giants on the Internet, cowards in real life” is used as an analogy to this idea, capturing a modern manifestation of the same paradox—individuals who exhibit dominance and aggression in virtual spaces yet display timidity and vulnerability in real-world contexts. It is not one of the hypotheses, but rather the central theme throughout the entire text.
Comments 3: 48: ‘the bullying things…’. Can you be more specific?
Response 3: Thank you for pointing this out. We agree with this comment. Since our focus is on whether individuals have experienced bullying, rather than specifying the exact types of bullying incidents, we have removed "the" and "things" and directly used the more concise phrase "experiencing bullying" to express our meaning more clearly.
Comments 4:50: I believe it should be ‘existing’ (not ‘existed’)
Response 4:Thank you for pointing this out. We agree with this comment. Therefore, we have changed “existed” to “existing”.
Comments 5:52: ‘being cybervictimisation…’ Can you please rephrase?
Response 5:Thank you for your valuable feedback. In response to your comment regarding the phrase "being cybervictimization," I have revised the sentence to make it more concise and clear. The revised sentence now reads: "Moreover, another study found that cybervictimization was strongly related to cyberbullying (Ak et al., 2015)."
Comments 6:53 and 54: linguistic corrections are necessary
Response 6:Thank you for pointing this out. We agree with your comment. We have made the following revision:
"Therefore, hypothesis 1 can be derived from the literature: there is a longitudinal relationship between real-life victimization and cyberbullying."
Comments 7:In 1.3., I did not find these surprising, nor did I find them something that can be described as specifically Chinese. There are similar idioms in other languages, which makes me think this might be more of a generalised perspective
Response 7:Thank you for pointing this out. The expressions“以暴制暴” (fight violence with violence) and “以毒攻毒” (use poison to fight poison) are not unique to Chinese culture; similar sayings exist in other languages as well. Our focus here is not on emphasizing their cultural specificity, but rather on highlighting the alternative perspective they offer for addressing negative behaviors. These cultural expressions provide an interesting viewpoint for considering non-traditional strategies to counter online bullying, which is worth further exploration.
Comments 8:I have some doubts as to the methodology and data collection. I believe the manuscript would benefit from some expansion, particularly in the reasons for selecting these age cohorts. Justifying this at this point would make it easier to properly identify the study's limitations and point out possibilities for further research at a later stage. What were the criteria for inclusion and exclusion? What were the questions asked? How were the questionnaires applied?
Response 8:We appreciate the reviewer’s valuable comments. We have expanded the Method section to provide a clearer justification for selecting the specific age cohort, detailed the inclusion and exclusion criteria, described the main questions and measures used, and clarified the procedure for administering the questionnaires. These additions improve transparency, allow for a better understanding of the study’s limitations, and highlight possibilities for future research.
Comments 9:Still on the same issue, why were there two phases (‘waves’?) for this study? Why was there such a long gap between them? Could you please expand on this? And why did 596 remain? Why were the others excluded?
Response 9:We appreciate the reviewer’s insightful questions. The study employed two waves of data collection in order to examine potential longitudinal relationships, which provide greater value than cross-sectional data. The relatively long interval between waves was chosen to capture meaningful changes over time, as a shorter interval might not allow sufficient observation of temporal dynamics. Regarding participant retention, 596 students remained in the second wave due to natural attrition, while the remaining participants were lost for reasons such as relocation or absence during data collection.
Comments 10:In 2.2.1., the date reference for the Olweus scale is missing here
Response 10: Thank you for pointing this out. We agree with that comment. We have made the following changes. I added the scale reference date, which is 2013.
Comments 11:141: lexical inconsistency: ‘contained’?
Response 11: Thank you for pointing this out. We agree with that comment. The specific modifications are as follows: "Moreover, age was included at T1, along with some other basic information."
Comments 12:The conclusion is relatively short. I believe it could be improved by either expanding it or incorporating in it the limitations and suggestions for further studies. If that were the case, the conclusion section could begin with this part, be followed by the limitations and, afterwards, would come the suggestions for further studies. This structure would enhance a sense of progression in the argumentation. As it stands, the conclusion sounds somewhat repetitive.
Response 12:We appreciate the reviewer’s valuable suggestion. The conclusion section has been expanded to provide a more comprehensive summary of the study, including the main findings, the study’s limitations, and suggestions for future research. This revised structure enhances the logical progression of the argumentation and strengthens the overall impact of the conclusion.
Comments 13:Comments on the Quality of English Language The manuscript requires a very thorough linguistic revision. There are syntactical and punctuation issues which sometimes hinder understanding. There are also some typos which should be addressed
Response 13:We sincerely appreciate the reviewer’s comments regarding the language quality. The manuscript has undergone a thorough linguistic revision, including corrections to syntax, punctuation, and typographical errors, to improve clarity, readability, and overall academic quality.

Round 2
Reviewer 1 Report
Comments and Suggestions for Authors
I fully appreciated the authors’ efforts to improve the overall quality of this work. In spite of that, I believe that the manuscript requires additional revisions. Below are my suggestions. I hope that they can improve the overall quality of this research.
First, I believe the authors should deepen the literature review in paragraph 1.1. and 1.2. seems to be drafted. In addition, the authors should explain the mechanism of moderation more clearly in paragraph 1.3. Moreover, the current study section fails to disclose the study’s logic, thereby failing to meet scientific standards. Furthermore, I have some concerns about the literature the authors reviewed. For instance, in the previous round of revisions, I suggested describing the works on the Dark Triad to explain its main features. However, I noticed that the reference to Giancola et al. (2023) is cited in the text, but is incorrect in the reference list. Please update the reference list to include the correct reference: Giancola, M., Palmiero, M., & D’Amico, S. (2023). The association between Dark Triad and pro-environmental behaviours: the moderating role of trait emotional intelligence (La asociación entre la Tríada Oscura y las conductas proambientales: el papel moderador de la inteligencia emocional rasgo). PsyEcology, 14(3), 338-362. In addition, to ensure conformity, please double-check all references in the text and in the reference list.
As for the Methods and Discussion sections, they addressed my previous comments. However, I believe the manuscript requires major revision, especially in the Introduction and the theoretical explanation of the study.
Author Response
Comments 1:
Furthermore, I have some concerns about the literature the authors reviewed. For instance, in the previous round of revisions, I suggested describing the works on the Dark Triad to explain its main features.
Response 1:Thank you for pointing this out. We agree with this comment. We have now refined the Introduction and Discussion sections to improve the theoretical background and ensure a more coherent linkage between the reviewed literature and the study’s hypotheses.
Comments 2:
However, I noticed that the reference to Giancola et al. (2023) is cited in the text, but is incorrect in the reference list. Please update the reference list to include the correct reference: Giancola, M., Palmiero, M., & D’Amico, S. (2023). The association between Dark Triad and pro-environmental behaviours: the moderating role of trait emotional intelligence (La asociación entre la Tríada Oscura y las conductas proambientales: el papel moderador de la inteligencia emocional rasgo). PsyEcology, 14(3), 338-362.
Response 2:Thank you for pointing this out. We agree with this comment. We have now added this article to the reference list and corrected the in-text citation accordingly.
Comments 3: In addition, to ensure conformity, please double-check all references in the text and in the reference list.
Response 3:We appreciate the reviewer’s helpful suggestion. Therefore, we have carefully checked all references in the text and the reference list, updated the reference list, and ensured the consistency and accuracy of the references.
Reviewer 2 Report
Comments and Suggestions for Authors
Please be sure to re-read this carefully or have someone else due so. Some changes need to be made. For example, you write "vic-tim ization" instead of victimization. Also, be sure you aren't relying on Chat GPT. While it may not be the case, often times when the - symbol is used within a sentence, it makes the professional reader question whether the writing is the authors' or from an AI assistant. This is important work, so I want to be sure it has no errors.
Author Response
Comments 1: Please be sure to re-read this carefully or have someone else due so. Some changes need to be made. For example, you write "vic-tim ization" instead of victimization. Also, be sure you aren't relying on Chat GPT. While it may not be the case, often times when the - symbol is used within a sentence, it makes the professional reader question whether the writing is the authors' or from an AI assistant. This is important work, so I want to be sure it has no errors.
Response 1:Thank you for pointing this out. We agree with this comment. Therefore, we have conducted a thorough manual review of the manuscript and corrected all grammatical and formatting errors, such as standardizing "vic-timization" to "victimization," to ensure the manuscript meets professional standards. We apologize for these oversights in the previous version and thank you for your valuable feedback in improving our manuscript.
Round 3
Reviewer 1 Report
Comments and Suggestions for Authors
The authors addressed all comments. However, please double check the DOI for all references.